# Evolution of Reproductive Life History in Mammals and the Associated Change of Functional Constraints

**DOI:** 10.3390/genes12050740

**Published:** 2021-05-14

**Authors:** Jiaqi Wu, Takahiro Yonezawa, Hirohisa Kishino

**Affiliations:** 1School of Life Science and Technology, Tokyo Institute of Technology, Meguro Ward, Tokyo 152-8550, Japan; 2Department of Molecular Life Science, Tokai University School of Medicine, Isehara, Kanagawa 259-1193, Japan; 3Faculty of Agriculture, Tokyo University of Agriculture, Atsugi City, Kanagawa 243-0034, Japan; cyclotis@gmail.com; 4Graduate School of Agricultural and Life Sciences, The University of Tokyo, Bunkyo Ward, Tokyo 113-8657, Japan; hirohisa.kishino@gmail.com; 5The Research Institute of Evolutionary Biology, Setagaya Ward, Tokyo, 138-0098, Japan

**Keywords:** rate-based prediction approach, evolutionary history of reproduction-related traits, gestation period, weaning period, time to independence, litter size, female sexual maturity age, reproductive strategy, genes identified for reproduction-related traits

## Abstract

Phylogenetic trees based on multiple genomic loci enable us to estimate the evolution of functional constraints that operate on genes based on lineage-specific fluctuation of the evolutionary rate at particular gene loci, “gene–branch interactions”. Using this information as predictors, our previous work inferred that the common ancestor of placental mammals was nocturnal, insectivorous, solitary, and bred seasonally. Here, we added seven new continuous traits including lifespan, bodyweight, and five reproduction-related traits and inferred the coevolution network of 14 core life history traits for 89 mammals. In this network, bodyweight and lifespan are not directly connected to each other; instead, their correlation is due to both of them coevolving with gestation period. Diurnal mammals are more likely to be monogamous than nocturnal mammals, while arboreal mammals tend to have a smaller litter size than terrestrial mammals. Coevolution between diet and the seasonal breeding behavior test shows that year-round breeding preceded the dietary change to omnivory, while seasonal breeding preceded the dietary change to carnivory. We also discuss the evolution of reproductive strategy of mammals. Genes selected as predictors were identified as well; for example, genes function as tumor suppressor were selected as predictors of weaning age.

## 1. Introduction

As fitness—how well an organism adapts to its environment—is described by the reproductive success of the individual, reproduction strategy is the most important life history trait for living species. For gametogony species, reproduction needs cooperation between the two sexes. The egg cell contains much more nutrition than the sperm cell; thus, from the beginning, the female’s contribution to the next generation is much greater than the male’s contribution. Mammalian species in particular are characterized by viviparity, and the female’s mammary glands produce milk for nursing her offspring. Therefore, parental investment in mammals covers a broad range including gestating and weaning the young, providing food, protecting the young from predators, huddling, grooming and carrying the offspring, and teaching the basic skills to survive. Such investments are mainly conducted by the female. Accordingly, they place physiological constraints on female mammals [1,2]. The heavy investment causes a smaller litter size, namely the number of offspring per litter, in mammals compared with reptiles and birds [3,4,5]. In some mammalian species, the male may also help to raise the young. Male provisioning in mammals may increase the litter size, indicating an evolutionary benefit in terms of reproductive success [6].

A trade-off between quantity and quality of offspring is a key reproduction strategy. As mentioned above, litter sizes of mammals are generally smaller than those of other vertebrates [5], suggesting that mammals are generally K-strategists, comprising species in which the populations are suitable for stability at or near the carrying capacity (K) of stable environments. The reproduction of K-strategists is usually slower and produces fewer offspring. Nevertheless, the numbers of offspring vary among mammalian species. For example, litter size of the common tenrec (*Tenrec ecaudatus*) ranges up to 32 (18 offspring per litter on average). Litter sizes are usually smaller in larger mammals, while the variance in litter sizes is larger in smaller mammals [5], suggesting that smaller mammals have various reproduction strategies. To clarify the coevolution pattern among reproduction-related life history traits is very important for understanding mammalian reproduction strategies. However, little is known about the transformations of reproduction strategies during mammalian evolution because the fossil record of reproduction-related traits is generally poorly preserved.

In our previous study [7], we built a new framework of statistical genomic analysis to reconstruct ancestral states. Selection pressure comes mainly from the environment. Biological traits including behavior are the result of adaptation to the environment. Therefore, fluctuations in the selective constraint on each gene may well record the history of biological traits. Based on this idea, we modelled the differences in the evolutionary rates of individual genes in the genome. In this model, genomic evolution is the context behind the evolution of each single gene. Each single gene has its own unique evolutionary rate (“gene effect”) due to its functional constraints. On the other hand, each species has its own genome-wide mutation rate (“genomic rate”, the background molecular evolutionary rates that are shared by all genes, see Wu et al., 2017 [7]), which depends on generation length and exposure to mutagens. Furthermore, each gene may have a species-specific acceleration or deceleration of the evolutionary rate, called “gene–branch interaction”. Evolutionary rates of individual genes in each species can be expressed by the product of these three rates, and the last one captures species-specific fluctuations of selective constraints from the environment.

We performed a regression analysis of the gene–branch interactions and traits of extant species and then used the fitted model to predict ancestral states. By this approach, we could reconstruct the evolutionary history of traits and obtained the genes that may be related to the trait. In our previous work, we mainly focused on discrete traits. By a simple modification, the approach can also be applied to continuous traits. Elucidating the evolutionary history, coevolution network, and genes related to these traits should provide a new view for understanding mammalian reproduction.

## 2. Method and Materials

### 2.1. Sequence Data and Species Tree

We downloaded alignments of 1204 single-copy genes of 89 mammals and the species tree in Wu et al., 2017 [7], and used them in this study.

### 2.2. Life History Traits Related to Animal Reproduction

Besides 10 life history traits published in Wu et al., 2017 [7], we collected seven additional traits that are related to reproduction of these 89 mammals, namely average gestation period, average weaning age, average time to independence, average number of offspring, female average age at sexual or reproductive maturity, lifespan, and bodyweight from the Animal Diversity Web (Appendix A).

### 2.3. Rate-Based Ancestral State Reconstruction on Continuous Traits

In Wu et al., 2017 [7], we modelled the rate of molecular evolution as the product of branch effect, gene effect, and gene–branch interactions. Branch effect depends on generation length and exposure to mutagens, and contains information on genomic evolution; gene effect is unique for each gene and is the level of functional constraints conducts on each gene; gene–branch interaction is the fluctuation of molecular evolutionary rate across each gene and branch and contains information on historical adaptations. By regressing gene–branch interactions to the trait value at terminal branches, we could obtain a model to predict trait values using gene–branch interactions. Applying the fitted model to gene–branch interactions of ancestral branches, we could then also predict ancestral states. The underlying assumption of this method is that traits with similar values also have a similar pattern of changes in molecular evolutionary rates of related genes. In Wu et al., 2017 [7], we mainly focused on discrete traits and used LASSO penalized logistic regression to fit the model; for continuous traits, we directly regressed gene–branch interactions to trait values. We used log-transferred trait values for continuous traits to fit the regression model. Ancestral states of all seven traits collected in this work were calculated and summarized in Appendix A. LASSO penalized logistic regression shrinks the coefficients of nonsignificant variables to zero, leaves only a small proportion of variables as significant predictors. A similar approach was used by Chikina’s group and associates the gene-specific rates with the change in trait states [8].

When we fit the model, we excluded terminal branches that have a missing value or uncertain trait states; for example, mammals that do not have any records on its weaning age were excluded when we fit the regression model of weaning age. The terminal branches that were removed during model-fitting also have a full set of gene–branch interactions. We could apply the fitted model to terminal branches and impute its trait values.

### 2.4. Trait Coevolution Network

Besides the seven traits newly collected in this work, we also included ten discrete traits that were published in Wu et al., 2017 [7]. We interpreted missing values using the regression model. We calculated partial correlations for ten discrete traits, five reproductive traits, lifespan and bodyweight using the R package ppcor [9]. To visualize the pattern of correlations among all traits, a minimum- Bayesian information criterion (BIC) graph were generated using the R package gRapHD [10]. Ancestral states obtained as mentioned above were used to create the trait coevolution network. We used igraph in R to draw the network [11]. 

We analyzed the coevolution between diet and seasonal breeding using the reversible-jump Markov Chain Monte Carlo model implanted in BayesTraits (V3.0.2) [12].

## 3. Results and Discussion

### 3.1. Correlations among Traits

In this work, besides lifespan and bodyweight, we collected five new reproduction-related traits including gestation period, weaning period, time to independence, litter size, and female sexual maturity age, all of which are continuous traits. We extended our rate-based ancestral prediction approach to continuous traits and reconstructed ancestral states of all five reproduction-related traits as well as lifespan and bodyweight.

How life history traits are correlated with each other is an area under active investigation and is one of the key questions for understanding mammalian evolution. Besides the seven continuous traits mentioned above, we analyzed 10 discrete traits including activity pattern (diurnal vs. nocturnal), sociality (social vs. solitary), diet, etc., in our previous work [7]. We calculated correlations among all 17 traits, and the correlations among reproductive life history traits are summarized in Appendix A. As the ancestral states of each trait are also used to calculate the correlations, the result is less biased due to phylogenetic inertia. Average gestation period is correlated with both lifespan and bodyweight. For lifespan, the correlation is as high as 0.726, and the correlation is 0.603 for bodyweight. As well as average gestation period, time to independence, weaning period, and female sexual maturity age are also correlated with lifespan, with values of 0.418, 0.518, and 0.485, respectively. The correlations of these reproduction-related traits with bodyweight show the same tendency (Appendix A). Since bigger animals are usually longer-lived, it is to be expected that they also show longer gestation and weaning periods. Litter size is negatively correlated with lifespan and bodyweight, with correlation values of −0.446 and −0.227, respectively. Notably, the correlations of these traits with lifespan are all larger than they are with bodyweight. Compared with bodyweight, lifespan is thus a more important factor for the respective traits.

### 3.2. Trait Coevolution Network

Reproductive life history traits influence each other. The pairwise correlation shown in Appendix A is possible due to an indirect correlation of traits with the third factors. Considering this, we further calculated partial correlation networks among traits. The edges shown in the figure are selected on minimum Bayesian information criterion scores (BIC), which indicate the coevolution relationships among traits.

We find that gestation period is most strongly connected to lifespan, with a positive partial correlation of 0.514. Time to independence is positively correlated with female sexual maturity age and weaning period, with partial correlation values of 0.411 and 0.325, respectively. A later time to independence of the young may increase the age of sexual maturity. Litter size is negatively correlated with gestation period and arboreality, with partial correlation values of −0.251 and −0.210, respectively. Longer gestation period and weaning period indicate heavier female investment in the offspring. When parental care becomes denser, litter size may become smaller, due to the trade-off in which reproductive investment constrains litter size [3]. Arboreal mammals generally have smaller litter sizes because it is difficult to carry many new babies after birth [13]. Diurnal mammals often have a longer weaning period than nocturnal mammals (correlation = 0.351), while monogamous mammals often have a shorter one (correlation = −0.172). There is also no direct relation between lifespan and bodyweight; their correlation is due to both factors being impacted by gestation period (correlation = 0.514 and 0.377, respectively). Male-biased sexual dimorphism (male-larger in Figure 1) is positively correlated with lifespan and weaning age, with correlation coefficients of 0.163 and 0.199, respectively. We did not find a direct correlation between the male-larger and monogamous traits. Seasonal breeding is positively correlated to omnivory (correlation = 0.169). The relation between diet change and reproductive seasonality is discussed in a later section as well. These finding provide new insights into how life history traits coevolve with each other.

### 3.3. Evolutionary History of Parental Investment in Mammals

Except for monotremes, viviparity is an important characteristic of mammals. After giving birth, female mammals spend time feeding milk to and rearing their offspring. For most species, females mainly contribute to the care of offspring, although male care also occurs in some cases. Life history traits that are related to parental investment in mammals are gestation period, weaning period, time to independence, and litter size. The length of the gestation period and weaning period mainly indicates how much the female contributes to the care of offspring, while time to independence and litter size are also impacted by male care [6]. As mentioned above, all of these traits are impacted to some extent by other life history traits, such as lifespan, bodyweight, and social behavior.

Figure 2 shows the evolutionary history of mother care (gestation period + weaning period), female sexual maturity age, and litter size. The common ancestor of placental mammals had a relatively short maternal care period, perhaps due to its short lifespan. However, before the Cretaceous–Paleogene (K–Pg) boundary, the average time for maternal care in ancestral mammals became longer. These ancestors, including the common ancestors of Euarchonta, Primatomorpha, Euungulata, Ferae, etc., show a prolonged duration of maternal care compared with the common ancestor of mammals. The results of time for female sexual maturity show a similar pattern, with some lineages displaying a prolonged time for female sexual maturity just before the K–Pg boundary. On the other hand, litter size shows a different pattern, namely that the ancestors of mammals generally show a small litter size compared with modern mammals. Collectively, these results indicate that mammals mainly use the K-strategy, which means a smaller litter size and more intensive care of offspring, to reproduce the next generation.

### 3.4. Coevolution between Diet and Seasonal Breeding

Reproduction of the next generation needs tremendous effort, and the requirement for food is the most important factor. For most of the species, the likelihood of successfully raising the young is highest during the season when food is most abundant. Food availability is considered one of the most important factors for seasonal breeding behavior [14]. Compared with omnivorous mammals, constraints on the availability of food are more severe for carnivorous and herbivorous mammals. Since omnivorous mammals have a wider selection of food that is less likely to be constrained by season, we expect that omnivorous mammals are more likely to reproduce year-round than carnivores and herbivores. The correlation between reproductive seasonality and omnivory is 0.237, which is much higher than that between reproductive seasonality and both carnivory (−0.181) and herbivory (−0.008).

We further tested the relationships between changing diet and the change between year-round (1) and seasonal (0) breeding of 89 mammals by the reversible-jump Markov Chain Monte Carlo model implanted in BayesTraits (V3.0.2) (Figure 3) [12]. The results are summarized in Figure 3. The change in diet and year-round breeding (1) to seasonal breeding (0) occur at the same rate for most pairs, especially for herbivores; however, there are two cases that show a different pattern. If the ancestors are year-round breeders, it is difficult to change their diet to a carnivorous one; similarly, if the ancestors are seasonal breeders, it is also difficult to change their diet to an omnivorous one. This result indicates that the availability of food is a prerequisite for seasonal breeding behavior but that, after the species has become fixed into a certain breeding type, either seasonal or year-round, seasonal breeding behavior may constrain their diet changes as well.

### 3.5. Genes Related to Reproductive Traits

The rate-based ancestral reconstruction approach regresses transformations of traits to fluctuations in the molecular evolutionary rates of genes. We used LASSO penalized regression to conduct the analysis; hence, genes that are not significant in predicting a trait have zero coefficients, leaving only genes that are significant in the prediction of trait values. Genes selected as predictors of the trait are likely related to the trait (Figure 4 and Appendix A). We found 23 genes as predictors of average weaning age. At least 11 tumor suppressor genes were selected, and some are related to, or function directly as, tumor suppressor genes in breast cancer (VCAN, CYLD, KLF10, CASS4, NUF2, DAAM2, PIP4K2A, PRKCD, etc.), mainly with a negative coefficient. VCAN, with the strongest coefficient of −0.484, is highly expressed in breast cancer progenitor cells [15]. Downregulation of CYLD (coefficient = −0.156) may promote breast cancer metastasis [16]; KLF10 (−0.151) is an anti-metastasis gene that significantly reduces breast cancer cell invasion [17]; low PIP4K2B (−0.047) expression in human breast tumors correlates with reduced patient survival [18]. NKTR, the natural killer cell-triggering receptor, is also selected. Longer weaning age is highly likely to be related to strengthened functional constraints on tumor suppressor genes. Genes related to average time to independence have multiple functions, such as immune system genes (NKTR, coefficient = −0.564), genes related to autism spectrum disorders (ERMN, −0.226) [19], and tumor suppressors (PIP4K2A, −0.059). The gene with the highest coefficient (−0.714) is IRAK3, which is associated with asthma [20]. Twelve genes were selected to be predictors of litter size. They function as a tumor suppressor (PIP4K2A, 0.039) [18], interfere with neuronal development and are related to autism (CSDE1, −0.039) [21], or function as a transcriptional repressor (ZBTB1, 0.126) [22]. Thirty-nine genes were selected to be related to average gestation period, and 76 were related to female sexual or reproductive maturity age. For many of these genes, genome-wide association studies indicate a relationship with body weight, body height, or BMI-adjusted waist–hip ratio (Appendix A). Notably, several genes were detected as predictors of several different traits, indicating the complex correlation structure and mechanisms of traits.

### 3.6. Evolution of the Mammalian Reproduction Strategy

Although litter size information is seldom preserved in fossil records in mammalian evolution, Hoffman and Rowe (2018) [11] reported a stem mammalian fossil (*Kayentatherium*: Mammaliaformes) from the Early Jurassic period in which the clutch comprises at least 38 perinates, which is outside the range of extant mammalian species. On the other hand, our ancestral state reconstruction of litter size (Figure 2) indicated that litter size in the common ancestor of Theria (Eutheria + Marsupial) in the Early Cretaceous period was small (2.7 individuals per litter). This finding suggests that the reproduction strategy of mammals shifted from r-selection to K-selection before the emergence of Theria. Interestingly, litter sizes of Theria had generally been small through the Cretaceous to the Paleogene periods, and litter sizes independently increased in the Neogene in multiple lineages (Figure 2). The transformation pattern of litter size as inferred from our rate-based analysis suggests that K-strategists served as the source of the diverse reproduction strategies in mammalian evolution. Stockley and Hobson (2016) [6] demonstrated the coevolution of paternal care and litter size. They suggested that the root of mammals had a small litter size without male provisioning and that the transition probability from “small litter size with male provisioning” to “large litter size with male provisioning” was higher than any other state changes. Their proposal is largely harmonious with our ancestral state reconstruction. The r-strategists are species that maximize reproductive capacity (r), generally reproduce faster, and produce larger numbers of offspring. The r-strategists typically prosper in unstable environments, and conversely the K-strategists generally occupy more stable environments. However, the fitness of r/K-strategists through mass extinctions has not been well documented. Extant mammals experienced at least two mass extinction events, at the Cretaceous–Paleogene boundary (66 Ma) and at the Eocene–Oligocene boundary (33 Ma) [7]. Considering that the K-strategists survived these two mass extinction events in our ancestral state reconstruction, it is possible that the fitness of mammalian K-strategists is higher in such periods.

Botha-Brink et al. (2015) [23] analyzed bone growth mark data of stem mammals around the Permian–Triassic boundary, known as the greatest mass extinction in Earth’s history, and demonstrated that late-breeding strategists generally became extinct whereas lineages that shifted to an early breeding strategy survived this mass extinction event. Late-breeding strategists re-emerged several million years later, in the Middle Triassic. Our ancestral state reconstruction on the sexual maturity age (Figure 2) also shows a compatible tendency. The Mesozoic mammalian species were generally late-breeding strategists, and they shifted to the early breeding strategy after the Cretaceous–Paleogene boundary.

From these findings, it is plausible that K-selected species with the early breeding strategy have the highest fitness to survive mass extinction events during mammalian evolution. The modern families of extant mammals mostly emerged after the global cooling event in the Oligocene period (33–23 Ma) [24,25]. Although it is possible that r-selected mammalian species repeatedly emerged in the Paleogene period (66–23 Ma), their lineages appear to have gone extinct or failed to shift their reproduction strategy. Mammalian r-strategists as seen in the Rodentia and Lagomorpha might have evolved new lineages that were adapted to the blank ecological niche after the mass extinction caused by the global cooling event in the Oligocene period.

## Figures and Tables

**Figure 1 genes-12-00740-f001:**
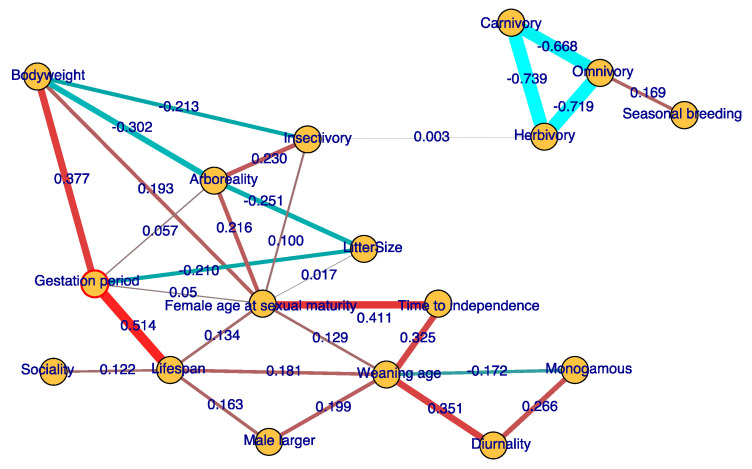
Coevolution network of life history traits. Based on the partial correlations of traits, we inferred a trait–coevolution network. The edges shown in the figure are selected on minimum Bayesian information criterion scores (BIC). Ancestral states of the traits are also used to infer the correlation network. Numbers along edges are partial correlation coefficients among traits, with the relative strength of the correlation indicated by the thickness of the edge and whether the correlation is positive or negative indicated by red–green color gradients.

**Figure 2 genes-12-00740-f002:**
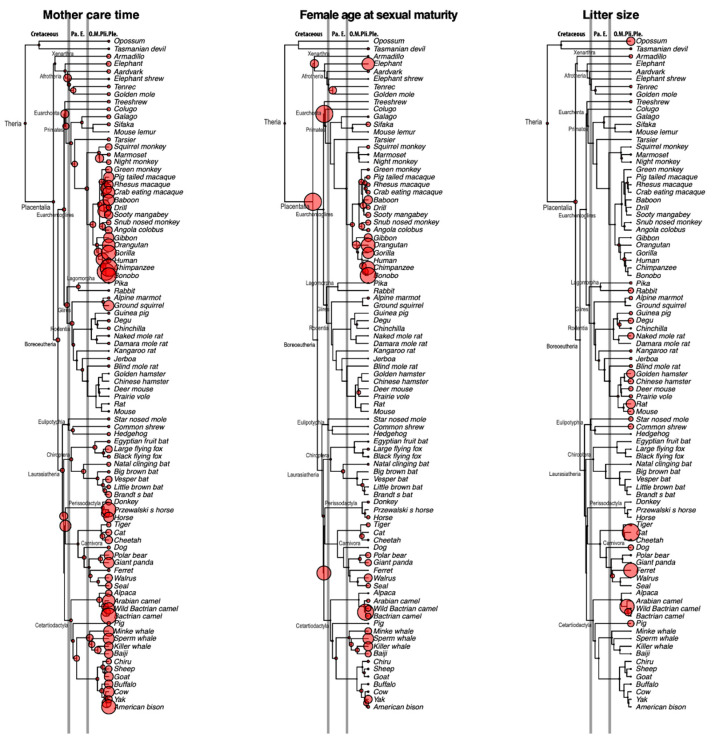
Evolutionary histories of mother care time, female age at sexual maturity, and litter size reconstructed by rate-based prediction. Nodal circle sizes are proportional to the magnitude of each quantitative trait. Absolute values of the ancestral traits are shown in Appendix A. Branch lengths are proportional to divergence times, as estimated in our previous study (Wu et al., 2017), and abbreviations of the geological epochs are shown at the top of tree (Pa: Paleocene, E: Eocene, O: Oligocene, M: Miocene, Pli: Pliocene, Ple: Pleistocene). Gray vertical lines indicate the Cretaceous–Paleogene boundary (66 Ma) and the Eocene–Oligocene boundary (33 Ma).

**Figure 3 genes-12-00740-f003:**
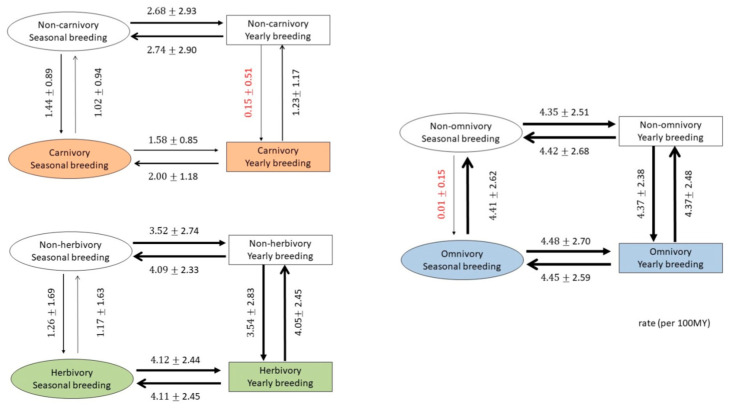
Coevolution between seasonal/yearly breeding and diet in mammals. Transitions can occur between four states: yearly reproduction and carnivory (left upper)/herbivory (left lower)/omnivory (right), seasonal reproduction and carnivory/herbivory/omnivory, yearly reproduction and non-carnivory/non-herbivory/non-omnivory, and seasonal reproduction and non-carnivory/non-herbivory/non-omnivory. Arrows indicating transitions between states are scaled to represent the probability of a transition: line thicknesses are proportional to transition rates. Absolute values of transition rates (per million years) are indicated next to each arrow.

**Figure 4 genes-12-00740-f004:**
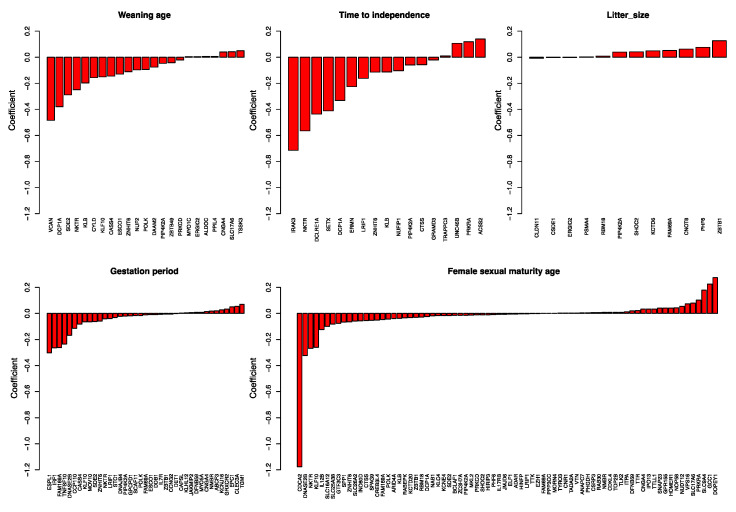
Coefficients of genes selected as predictors of traits. Genes selected as predictors of weaning age, time to independence, litter size, gestation period, and female sexual maturity age are listed, with their coefficients shown as bar plots.

## Data Availability

Correspondences and requests for materials should be addressed to wujiaqi06@gmail.com.

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
