# Peer review of "Evolution of Reproductive Life History in Mammals and the Associated Change of Functional Constraints"

_genes, 2021, doi:10.3390/genes12050740_

Round 1

Reviewer 1 Report

The authors analyzed relationships between a phylogenetic tree and quantitative trait information of mammals. It might be interesting to detect the traits' co-evolutionary network and extract genes-related reproductive traits are also related tumor if they are true. This manuscript would complement the authors' previous paper, which focused on a large-scale analysis of mammalian phylogeny and more traits. However, some tables (Table 1, Table S1, and S3) and methods are missing from the manuscript, and some concerns should be clear before publication. 

#1  Line 97. 
Table S1 is a correlation matrix but not the original data of Animal Diversity Web. Please insert the correct data and URL of the source.

#2 Line 147. Table 1 is missing. Please insert it. 

#3 Most traits in Fig1 are reasonable and commonly known, but I'm not sure "genomic rate" is a common term; it seems to be an original word proposed by the authors in the previous paper (Wu et al., 2017). It is unbalanced because the authors define "litter size" as above, but it lacks a definition of genomic rate. Please explain it in this manuscript. 

More importantly, genomic rate is not a trait (phenotype). Only genomic rate is not a trait in the network. I feel it's strange a little bit. 

#4 Line 155 and Fig1. In the coevolution network of life-history traits, seasonal breeding and male larger are isolated from other traits. Why? Are there any evidence or suggestions from the previous studies?

#5.  In section 3.4, the authors tested relationships between changing diet and year-round/seasonal breeding. However, it conflicts with the authors' explanation "Because traits related to diet have strong negative correlation with each other, due to the way we define the trait state (e.g., carnivorous or not), we did not include traits related to diet in the current network. "  

I looked at Table S1; it is also true for other traits. For example, litter size showed only four positive correlations with other traits (genomic rate, seasonal breeding, Monogamous, Omnivory.), while Herbivory has six positive correlations. It might be cherry-picking. The authors should include diet traits in the network if the authors test relationships between diet-breeding.

#6  The correlation between seasonal-breeding-omnivory is only 0.237. It is pretty low, even if the other relationships (seasonal-breeding-carnivory) are much lower than this. Why is it so low in mammals? Is the state transition between diet and breeding styles flexible in mammals? Is there any idea to explain it in the context of mammalogy?

#7. Methodologically, the authors use a regression model to infer many missing values of traits. This imputation method may contribute to lower correlations. Please explain more details of the imputation in the method section.

#8. In section 3.5, the Materials and Methods section lacks a description of how the authors extract genes related to a tumor. 

I randomly checked some genes in Table S3 and the manuscript. First, the authors listed some tumor suppressor genes in breast cancer in the maintext, line 255.The gene card did not suggest VCAN is related to breast cancer, but another disease, Wagner syndrome. https://www.genecards.org/cgi-bin/carddisp.pl?gene=VCAN

Another example, KLB is also noted as a " tumor suppressor in human breast cancer," and the info was from gene card according to Table S3. However, opening a gene card, there is no description about "tumor" or "cancer." https://www.genecards.org/cgi-bin/carddisp.pl?gene=KLB

I doubt the list is not correctly accumulated. Instead, I suggest a systematic analysis (like GO analysis) might be suitable for this purpose. 

Minor points

1. Line 148. The authors explain "litter size, namely the number of offspring per litter" in the Results section. However, litter size appears in the Introduction. Please explain it in the Introduction.

2. Line 219. Please check the citation (12). Ref12 is a cancer paper and does not match the explanation of seasonal breeding in the line. 

3. Line 225. Please suggest original data of the correlations, such as carnivory (0.181) and herbivory (-0.008), perhaps from Supplementary data?

4. Line 248. Legend for Fig 3 is strange. "~I think" should be removed or replaced in the figure legend.  

5.  It seems that Table S3 is missing in the main text. 

Author Response

Thank you very much for reviewing our work. We thank your questions and comments very much and think they largely helped us to improve our manuscript. We hope our revised manuscript and tables will make you feel satisfied. Please find the reponse to reviewer letter in attached file.

Reviewer 2 Report

The topic presented is interesting and the paper is well written and readable. Data treatment and the statistical analysis (regression) are sound. The authors study evolutionary continuous traits. The approach presented is based on genomic data of 89 different mammals and the coevolution network of life history traits. They also use the Monte-Carlo method for a Markov chain model. The goal is to better understand the mammalian reproduction from the evolutionary point of view.

I recommend this paper for publication by the Journal. I strongly suggest to add the following sentence, with two new references, at the end of the paragraph in line 80: “For the study of optimal sex-ratio and the evolution of sex-reversal in age-structured mathematical models, see the papers [*] and [**], where the authors derive specific fitnesses based on vital rates of the populations and their interactions with the environment.”

[*] Mimmo Iannelli and Jordi Ripoll. Two-sex age structured dynamics in a fixed sex-ratio population. Nonlinear Analysis: Real World Applications, 13 (2012) 2562--2577.

[**] Àngel Calsina and Jordi Ripoll. Evolution of age-dependent sex-reversal under adaptive dynamics. J. Math. Biol., 60(2):161--188, 2010.

Author Response

Thank you very much for reviewing our work. Please find the reponse to reviewer letter in attached file.

Reviewer 3 Report

In this study, Wu and coworkers, by analyzing evolutionary rates of a large array of genes in a phylogeny of 89 mammals, extend their previous analysis (Curr Biol 2017 9;27:3025-3033.e5. doi: 10.1016/j.cub.2017.08.043.  Rates of Molecular Evolution Suggest Natural History of Life History Traits and a Post-K-Pg Nocturnal Bottleneck of Placentals) to the coevolution of continuous traits related to lifespan, body weight and reproduction.

The paper is interesting and pleasant to read and the new analysis presented extends the previous insight of their Curr. Biol. Paper.

Many imprecisions are present and the paper needs carful editing before publishing. For example:

The key reference Wu et al. 2017 has a wrong numbering (7 instead of 6…)

Other references are imprecisely cited

In the introduction line 44 “he female’s” is strange…

Line 45 “although” should be eliminated

Line 69 "are" instead of "is"

Line 184 “Expect” is wrong “Except…”

These are just few examples; a good re-reading is necessary and possibly the input of a native English speaker.

Author Response

Thank you very much for reviewing our work. We further checked our English and hope this time, it is improved. Please find the reponse to reviewer letter in attached file.
